# Incidence and Risk Factors for Progression to Diabetes Mellitus: A Retrospective Cohort Study

**DOI:** 10.3390/ijerph19010123

**Published:** 2021-12-23

**Authors:** Min Kyung Hyun, Jong Heon Park, Kyoung Hoon Kim, Soon-Ki Ahn, Seon Mi Ji

**Affiliations:** 1Department of Preventive Medicine, College of Korean Medicine, Dongguk University, Gyeongju 38066, Korea; mk3three@dongguk.ac.kr; 2National Health Insurance Service, Wonju 26464, Korea; parkjh@nhis.or.kr; 3Health Insurance Review & Assessment Service, Wonju 26465, Korea; rudgns112@hira.or.kr; 4Public Health and Medical Services Office, Chungnam National University Hospital, Daejeon 35015, Korea; withspirit09@gmail.com

**Keywords:** diabetes mellitus, impaired fasting glucose, incidence, risk factors

## Abstract

(1) Objective: This study examined the incidence and risk factors contributing to the progression to diabetes mellitus (DM) in a seven-year follow-up study of non-diabetic National Health Examinees. (2) Methods: For this retrospective observational cohort study, we used two national representative databases: the National Health Screening (HEALS) database 2009 and the National Health Insurance Service (NHIS) database 2009–2015. The eligible subjects without DM with blood sugar levels of <126 mg/dL were selected using the HEALS database. The subsequent follow-up and clinical outcomes were evaluated using the NHIS database. Cox proportional hazard regression was applied to examine the effects of the covariates on progression to diabetes. (3) Results: Among those who took part in the national health screening in 2009, 4,205,006 subjects who met the eligibility criteria were selected. Of these, 587,015 were diagnosed with DM during the follow-up by 2015. The incidence of progression from non-diabetes to DM was 14.0%, whereas that from impaired fasting glucose (IFG) to DM was 21.9%. Compared to the normal group, the newly diagnosed DM group was more likely to comprise older, female, currently smoking, and high-risk drinking participants and participants with IFG, hypertension, dyslipidemia, and metabolic syndrome. (4) Conclusions: This epidemiological study in the Republic of Korea found risk factors similar to those of other studies, but the incidence of progression to DM was 22.8 per 1000 person-years, which is higher than that previously reported. Hence, more care is needed to prevent DM.

## 1. Introduction

Diabetes mellitus (DM) is one of the world’s leading public health concerns; it has increased steadily in incidence over the past few decades and resulted in complications in multiple organ systems [1]. The number of people with DM has risen from 285 million in 2009 to 463 million in 2019 and is expected to increase to 578 million by 2030 [1].

Prediabetes is defined when the glucose levels do not meet the criteria for DM but are too high to be considered normal, including impaired fasting glucose (IFG) and impaired glucose tolerance (IGT) [2]. Among them, IFG is diagnosed when the blood glucose levels are between 100 and 125 mg/dL (5.6 to 7.0 mmol/L), and it is highly associated with the development of DM [2]. IFG increases the risk of progression to DM, and the annual rate of conversion to DM is 6–9% [3,4,5]. The prevalence of IFG ranges from 1% to 30%, depending on the country and its estimated standards [6,7,8]. According to the American Diabetes Association (ADA), the risk factors for DM are age, gender, obesity, family history, physical inactivity, hypertension, and prediabetes including IFG [3]. On the other hand, the incidence and risk factors for chronic diseases, such as DM, can also show interregional, ethnic, and racial differences [9]. Indeed, some studies reported that Asian people have a higher risk of developing diabetes than other ethnicities, as well as a higher incidence [10,11]. A Japanese epidemiological study reported that IFG increased the risk of diabetes 8.8-fold [12]. However, few epidemiological studies on Koreans derived from a representative large-scale database that complements the strengths and weaknesses of each dataset have been reported.

Knowledge of the exact incidence and prevalence of prediabetes and diabetes is essential for effectively managing diabetes care and health insurance finances. In particular, the incidence of progression and risk factors of progression from non-diabetes, including IFG, to DM need to be determined.

In the present study, the subjects screened for normal glucose and IFG in the 2009 National Health Screening (HEALS) were followed up for seven years using the National Health Insurance Service (NHIS) database to investigate the incidence and risk factors of DM.

## 2. Materials and Methods

### 2.1. Study Design and Data Source

This was a retrospective observational cohort study using two national representative databases: the HEALS and NHIS databases owned by the NHIS system, a single insurer in the Republic of Korea (ROK) (Figure 1) [13,14]. The NHIS data cover approximately 96.6% of the ROK population and include the demographic and medical treatment information of the participants [13]. In contrast, HEALS data represent a cohort of people who participated in national health screening programs provided by the NHIS. It contains information on the health problems and risk factors of examinees obtained through the national health screening programs [14]. The HEALS and NHIS databases are linked to personal identification numbers, but the data are anonymized and provided to a designated secure computer in a security room. These data can be used to identify the incidence of new-onset DM among non-diabetic individuals at a health screen event and discover the risk factors for new-onset DM in the ROK.

### 2.2. Study Subjects and Setting

The eligible target subjects were selected using the HEALS database and subsequently followed up until 2015 when access was granted for analysis. The clinical outcomes were checked using the NHIS database (Figure 1).

Briefly, 10,644,911 adults participated in HEALS in 2009. The exclusion criteria were as follows: (i) initial blood sugar values of ≥126 mg/dL, (ii) diagnosis of DM (ICD-10-CM: E10, 11, 13, and 14), (iii) history of malignancy (International Classification of Diseases, Tenth Revision, Clinical Modification (ICD-10-CM): C00 to 97), and (iv) missing data in the variables including smoking/drinking/exercise. Finally, 4,205,006 target subjects were selected for the study (Figure 2).

### 2.3. Variables

#### 2.3.1. Outcome Variables

The primary clinical endpoint was the progression from non-diabetes to DM, and new-onset DM was defined as a diagnosis with DM (ICD-10-CM: E10, 11, 13, 14) from the medical records of the NHIS database. The diagnostic criteria for DM in ROK are based on a fasting plasma glucose of ≥126 mg/dL for eight hours, or a two-hour plasma glucose of ≥200 mg/dL during a 75 g oral glucose tolerance test, or a glycosylated hemoglobin (A1C) level of ≥6.5% [15]. Repeated verification is required on another day if there were no obvious hyperglycemia symptoms (polyurea, polydipsia, and unexplained weight loss), but two or more of the above abnormal results from the same sample can be provided immediate confirmation [15].

#### 2.3.2. Household Income

The household income was calculated based on the insurance owner’s income level to claim health insurance premiums and was classified into quintiles. The higher the quintile, the higher the income level.

#### 2.3.3. Metabolic Syndrome

Metabolic syndrome was defined in individuals meeting three or more of the following criteria: (1) abdominal obesity, with waist circumference of ≥90 cm in men or ≥85 cm in women; (2) hypertriglyceridemia, with triglyceride (TG) of ≥150 mg/dL or medication use; (3) low high-density lipoprotein (HDL)–cholesterol, with HDL-cholesterol of <40 mg/dL in men and <50 mg/dL in women; (4) high systolic blood pressure (BP), with systolic BP of ≥130 mmHg and/or diastolic BP of ≥85 mmHg; or (5) hyperglycemia, with Fasting Plasma Glucose (FPG) of >100 mg/dL or medication use [16].

#### 2.3.4. Current Smoking, High-Risk Drinking, and Proper Exercise

Current smoking was defined in those who had smoked more than 100 cigarettes in their lives or were currently smoking [17].

High-risk drinking was defined as drinking more than 300 mL of alcoholic beverages per day on average. For traditional Korean drinks, one standard drink unit corresponds to one bowl (approximately 300 mL) of Korean rice beer (Makgeoli) or a quarter bottle (approximately 90 mL) of 20% Korean liquor (Soju) [17,18].

Proper exercise was defined as follows: (1) intensive exercise lasting more than 20 min per session and more than three times per week or (2) moderate exercise lasting more than 30 min per session and more than five times per week [19,20].

### 2.4. Statistical Analyses

The continuous variables are presented as the mean and standard deviation (SD) and were compared using Student’s t-test. The categorical variables are presented as a proportion and were compared using a chi-square test. The relationships between the dependent variable (progression to DM) and the various risk factors or independent variables were examined via the Cox proportional hazard model. Multicollinearity analysis with the variance inflation factor (VIF) was performed to identify the collinearity between the variables. Variables with VIF > 5 were considered to show severe multicollinearity; there were no variables with VIF > 5 in the model. The Cox proportional hazard model was applied to consider the timing of the event. The onset of DM was defined as an event and was censored when the follow-up was terminated or death occurred. In other words, the Cox proportional hazard regression estimated the prognostic influence of the non-diabetes status on the conversion to DM, while simultaneously controlling for the confounding effects of covariates. This model estimated the instantaneous relative risk of conversion to DM, averaged over the entire follow-up duration. The proportional hazard assumption was tested using the goodness-of-fit test, which compares the observed and expected risk probabilities. The adjusted hazard ratios (HRs) and 95% CIs are reported. Subgroup analysis was conducted on participants over 40 years of age because the prevalence of DM in Koreans in 2016 exceeded 10% for men in their 40s and women in their 50s. The data were analyzed using SAS statistical software, version 9.4, for Windows (SAS, Cary, NC, USA); two-sided probability values less than 0.05 were considered significant.

## 3. Results

### 3.1. Baseline Demographics

Of the 4,205,006 participants analyzed, the mean age was 40.1 ± 12.2 years, with 71.8% being male. This is because males are more likely to participate in the national health screening [21]. The proportion of participants with IFG with fasting blood sugar levels of 100–125 mg/dL was 24.4%, of which 80.6% were male. Metabolic syndrome was detected in 60.9% of the total: 64.0% males and 36.0% females. The proportions of all variables were significantly higher in males than in females (*p* < 0.001) (Table 1).

### 3.2. Incidence and Characteristics of Progression from Non-Diabetes to Diabetes

The cumulative incidence of DM was 14% for the seven-year follow-up, and the conversion rate of non-diabetes to DM was 22.8 per 1000 person-years (Table 2). The mean age of the 587,015 new-onset DM cases was 48.0 ± 12.7 years, which was higher than the 38.8 ± 11.6 years in the patients not diagnosed with DM; 21.9% of IFG subjects were diagnosed with DM compared to 11.4% of subjects with normal blood sugar levels; and 33.7% of subjects with hypertension were diagnosed with DM, which was higher than the 12.4% of non-hypertensive cases (Table 3).

### 3.3. Risk Factors of Progression from Non-Diabetes to Diabetes

The newly diagnosed DM group were characterized by IFG (HR 1.552, 95% CI: 1.542–1.563), older age (if 70 and above, HR 7.504, 95% CI: 7.389–7.620), female sex (HR 1.198, 95% CI: 1.190–1.206), high household income (if fifth quintile, HR 0.931, 95% CI: 0.924–0.939), hypertension (HR 1.433, 95% CI: 1.433–1.454), dyslipidemia (HR 1.256, 95% CI: 1.239–1.272), high triglyceride (HR 1.203, 95% CI: 1.195–1.212), high BMI (HR 1.048, 95% CI: 1.046–1.049), metabolic syndrome (HR 1040, 95% CI: 1.030–1.049), current smoking (HR 1082, 95% CI: 1.075–1.088), and high-risk drinking (HR 1140, 95% CI: 1.132–1.148), as compared to the normal group (Table 4). The results were similar in the analysis of participants aged 40 years and over (Table 5).

## 4. Discussion

The incidence of and risk factors contributing to DM progression in a follow-up study of non-diabetic national health examinees were examined by linking two types of national representative databases in the ROK. Approximately 14% of people not diagnosed with DM in the 2009 National Health Screening were diagnosed with DM over the seven-year follow-up. Furthermore, 24.4% of all subjects had IFG at the 2009 screening; 21.9% of those had converted to DM by 2015. In addition, the conversion rate from non-diabetes to DM was 22.8 per 1000 person-years.

A study of the 40~69-year age group without DM at the baseline examination in 2001~2002 using the Korean Genome and Epidemiology Study (KoGES) revealed an overall DM incidence of 22.1 per 1000 person-years after a 12-year follow-up [22]. The follow-up period was longer than that of the present study, and the study subjects were limited to residents of certain regions over 40 years of age. On the other hand, the conversion rate was 22.1, which is slightly lower than that in the present findings. These results may be because the data was collected in 2001—much earlier than 2009, the collection year of the present study. According to the NHIS database, the number of DM patients has increased steadily from 2.7 million in 2016 to 3.3 million in 2020. Of these, 95% of patients with DM in 2020 were in their 40s or older [23].

According to a previous study, among the 6.4 million members of the Hong Kong population who used hospital authority services from 2006 to 2014, the incidence of DM was 5.9% (*n* = 377,565), and the conversion rate to DM was 9.46 per 1000 person-years in 2014, which is much lower than that in the present findings [24]. In relatively old data from Japan, a systematic review and meta-analysis of the studies conducted between 1980 and 2003 resulted in a pooled DM incidence rate of 8.8 (95% confidence interval, 7.4–10.4) per 1000 person-years [25]. On the other hand, the Chennai Urban Rural Epidemiology Study (CURES) cohort (*n* = 1376), which followed an Asian Indian cohort for 9.1 years until 2013, reported a 30% incidence of DM [10]. In addition, they reported DM conversion rates of 33.1 per 1000 person-years in non-diabetics, including those with prediabetes, and 61.0 per 1000 person-years in IFG subjects [10]. In this study, female subjects exhibited 1.01 times higher conversion from non-diabetes to DM and 1.10 times higher conversion from IFG/IGT to DM than male subjects, but this difference was not statistically significant [10]. Similarily, the conversion to DM was 1.521 times higher in IFG subjects than in normal-glucose subjects, and the conversion to DM in non-diabetics was 1.198 times higher in females than in males [10]. On the other hand, the incidence of DM and the conversion rate of DM were overwhelmingly higher than those of the present research. At the start of the observation, the proportion of IFG subjects was 4.9% (while the proportion of all types of prediabetic subjects was 21.7%), which is lower than the 24.4% of IFG in the present study [10]. These differences may be related to the metabolic effects of the western-style diet, or tissue resistance to insulin. One study reported that Asian Indian people have the highest incidence of DM when compared to other Asian people [26]. The Southall And Brent Revisited (SABRE) study (*n* = 1007) that observed South Asian men 40–69 years of age living in North and West London for 19 years until 2011 reported a 35% incidence of DM [11]. This is higher than the 14.1% incidence of DM in men in the present study. Therefore, Asian ethnic groups have different incidence rates of DM, depending on their growth environment.Accordingly, more epidemiological studies are needed under a range of conditions to better understand the status and trends, suppress the increase in the incidence of prediabetes and DM, and set healthcare policies for prevention and treatment that are compatible with the burden of prediabetes and DM.

In the present study and reported research, IFG is a risk factor of progression to DM. A systematic review by the U.S. Preventive Services Task Force (USPSTF) reported that the treatment of IFG is associated with delayed progression to DM [27]. Therefore, it is necessary to prevent progression to DM in subjects with IFG by applying the appropriate lifestyle and medical interventions [2]. In particular, subjects with IFG are at higher risk of developing cardiovascular disease. Therefore, they require intensive cardiovascular risk management [28]. The National Institute for Health and Care Excellence (NICE) suggested a national strategy and policy to prevent DM linked to diet, physical activity, and obesity. In addition, adult DM patients need to manage their blood pressure, lipids and cardiovascular risk, blood glucose, and complications [29]. In line with these recommendations, hypertension, dyslipidemia, higher BMI, and metabolic syndrome were also risk factors for DM in the present study. However, there is also a report that IGT is not a risk factor for conversion to diabetes, which may explain partly why some subjects without metabolic syndrome converted to diabetes in present study [30].

Health behaviors are also linked to the development of DM. Smoking increases the risk of DM by affecting visceral abdominal fat accumulation, insulin resistance, and pancreatic b-cell dysfunction [31,32]. Various epidemiological studies have demonstrated a risk of DM and its complications linked to smoking [33,34,35,36]. Alcohol consumption also increases the risk of DM and its complications by affecting plasma glucose, gluconeogenesis, and insulin sensitivity [37]. In contrast, physical activity decreases the risk of DM and its complications [38,39,40]. In the present study, proper exercise was not a significant factor in the incidence of DM, which appears to be due to the limitation that it is not an accurate measurement of the amount and quality of exercise. Unlike smoking and drinking, exercise may have behavioral variations in terms of continuity and addictive behavior. Therefore, repeated measurement data using accurate tools will be needed. For example, a Korean study measured physical activity using a self-reported international physical activity questionnaire and reported a lower trend in the incidence of DM [38]. Unlike the incidence and conversion rates of DM, racial and regional differences in the risk factors appear to be indistinguishable.

Care is needed to prevent DM, and more efforts will be needed to reduce the risk of DM and the number of diabetic patients. In addition, more epidemiological studies in DM are needed to identify and alleviate the disease burden of DM and its complications.

This study had some limitations. First, the HEALS and NHIS databases are secondary databases that were not planned and collected according to specific research objectives, thereby limiting the validity of variables’ definitions and research results. These secondary data used limited by the lack of detailed clinical information to study a specific disease due to the data used for insurance claims. For example, the German Diabetes Association and the German Clinical Chemistry Association, in their 2019 guidelines, recommend caution using HbA1c to diagnose diabetes in the elderly [41]. Nevertheless, it is difficult to discuss these issues in the data used. It may be that Korean doctors diagnosed and entered the diagnosis code according to the diabetes diagnostic criteria recommended by the latest diabetes guidelines. On the other hand, the use of secondary databases can include large-scale groups of research participants, which makes it easier to generalize the results and reduces selective reporting. Second, this studyexcluded subjects without information on smoking, drinking, and exercise in the subject selection process. Such exclusion was an inevitable choice because health behavior information is important for finding the risk factors for the conversion to DM. Third, the changes in health behaviors during the follow-up period could not be analyzed due to data limitations. Despite these limitations, these findings provide comprehensive epidemiological information on diagnosed IFG and DM in the ROK using a large national population-based sample.

## 5. Conclusions

This epidemiological study in the ROK, which was a seven-year follow-up of non-diabetics from National Health Examinees, showed that the risk factors are similar to those for other regions and races. The incidence of progression to DM was 22.8 per 1000 person-years, which is higher than that found in previous studies in the ROK, but much lower than the incidence reported in an Asian Indian population.

## Figures and Tables

**Figure 1 ijerph-19-00123-f001:**
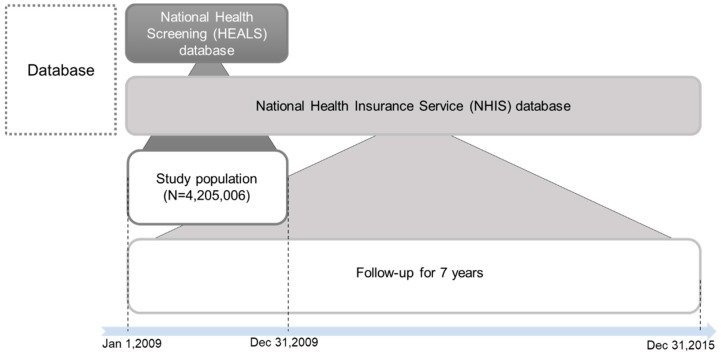
Study design.

**Figure 2 ijerph-19-00123-f002:**
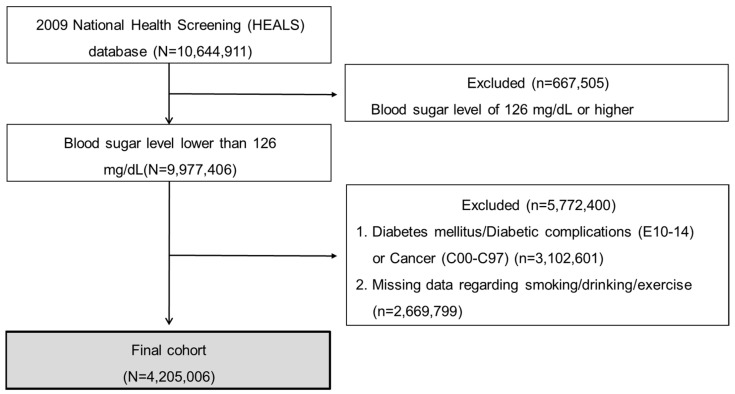
Selection of the study subjects.

**Table 1 ijerph-19-00123-t001:** Characteristics of the participants.

Variables		Total		Male		Female		*p*-Value
		*n*	%	*n*	%	*n*	%	
Total (*n*)		4,205,006	100.0	3,019,103	71.8%	1,185,903	28.2%	
IFG	No	3,179,247	75.6%	2,192,290	69.0%	986,957	31.0%	<0.0001
	Yes	1,025,759	24.4%	826,813	80.6%	198,946	19.4%	
Age group		40.1 ± 12.2		40.4 ± 11.9		39.3 ± 12.8		<0.0001
	18~29	977,664	23.3%	615,826	63.0%	361,838	37.0%	<0.0001
	30~39	1,254,309	29.8%	999,030	79.6%	255,279	20.4%	
	40~49	1,078,746	25.7%	754,509	69.9%	324,237	30.1%	
	50~59	585,787	13.9%	421,115	71.9%	164,672	28.1%	
	60~69	237,616	5.7%	177,846	74.8%	59,770	25.2%	
	≥70	70,884	1.7%	50,777	1.68	20,107	28.4%	
Household income	1	647,627	15.4%	400,233	61.8%	247,394	38.2%	<0.0001
	2	747,199	17.8%	455,816	61.0%	291,383	39.0%	
	3	928,682	22.1%	669,523	72.1%	259,159	27.9%	
	4	989,910	23.5%	775,989	78.4%	213,921	21.6%	
	5	891,588	21.2%	717,542	80.5%	174,046	19.5%	
Hypertension	No	3,901,156	92.8%	2,784,454	71.4%	1,116,702	28.6%	<0.0001
	Yes	303,850	7.2%	234,649	77.2%	69,201	22.8%	
Systolic blood pressure	<120	1,720,188	40.9%	1,012,586	58.9%	707,602	41.1%	<0.0001
	120~139	2,083,991	49.6%	1,673,166	80.3%	410,825	19.7%	
	≥140	400,827	9.5%	333,351	83.2%	67,476	16.8%	
Diastolic blood pressure	<80	2,279,381	54.2%	1,441,408	63.2%	837,973	36.8%	<0.0001
	80~89	1,522,799	36.2%	1,236,603	81.2%	286,196	18.8%	
	≥90	402,826	9.6%	341,092	84.7%	61,734	15.3%	
Dyslipidemia	No	4,108,644	97.7%	2,939,918	71.6%	1,168,726	28.4%	<0.0001
	Yes	96,362	2.3%	79,185	82.2%	17,177	17.8%	
Triglyceride	<150	2,926,610	69.6%	1,887,318	64.5%	1,039,292	35.5%	<0.0001
	≥150	1,278,396	30.4%	1,131,785	88.5%	146,611	11.5%	
HDL	M < 40, F < 50	594,305	14.1%	362,835	61.1%	231,470	38.9%	<0.0001
	M ≥ 40, F ≥ 50	3,610,701	85.9%	2,656,268	73.6%	954,433	26.4%	
BMI		23.6 ± 3.2		24.1 ± 3.0		22.4 ± 3.2		<0.0001
Waist circumference	M < 90, F < 85	3,524,867	83.8%	2,461,292	69.8%	1,063,575	30.2%	<0.0001
	M ≥ 90, F ≥ 85	680,139	16.2%	557,811	82.0%	122,328	18.0%	
Metabolic syndrome	No	1,643,100	39.1%	1,380,427	84.0%	262,673	16.0%	<0.0001
	Yes	2,561,906	60.9%	1,638,676	64.0%	923,230	36.0%	
Current smoking	No	2,540,898	60.4%	1,427,760	56.2%	1,113,138	43.8%	<0.0001
	Yes	1,664,108	39.6%	1,591,343	95.6%	72,765	4.4%	
High-risk drinking	No	3,645,544	86.7%	2,495,166	68.4%	1,150,378	31.6%	<0.0001
	Yes	559,462	13.3%	523,937	93.7%	35,525	6.3%	
Proper exercise	No	3,452,613	82.1%	2,436,072	70.6%	1,016,541	29.4%	<0.0001
	Yes	752,393	17.9%	583,031	77.5%	169,362	22.5%	

Abbreviations: IFG, impaired fasting glucose; HDL, high-density lipoprotein; BMI, body mass index.

**Table 2 ijerph-19-00123-t002:** Incidence of progression from non-diabetes to diabetes.

Follow-Up Period	Follow-Up Population	Newly Diagnosed Diabetes	Censored ^1^	Cumulative Incidence of Diabetes		Person-Years ^2^	Conversion Rate ^3^
Year	N	*n*	*n*	*n*	%		
<1	4,205,006	55,129	4,165	55,129	1.3%	4,202,924	13.1
1~2	4,145,712	88,029	4,881	143,158	3.4%	4,143,272	21.2
2~3	4,052,802	93,463	5,259	236,621	5.6%	4,050,173	23.1
3~4	3,954,080	95,179	5,286	331,800	7.9%	3,951,437	24.1
4~5	3,853,615	102,731	4,910	434,531	10.3%	3,851,160	26.7
5~6	3,745,974	104,789	4,321	539,320	12.8%	3,743,814	28.0
6~7	3,636,864	47,695	3,589,169	587,015	14.0%	1,842,280	25.9
Total	4,205,006			587,015	14.0%	25,785,058	22.8

^1^ If the follow-up was terminated or death occurred; ^2^ Censored times half-year assumption (follow-up population − 0.5 × Censored); ^3^ Number of diabetes progressions from non-diabetes per 1000 person-years.

**Table 3 ijerph-19-00123-t003:** Characteristics of the progression from non-diabetes to diabetes.

Variables		Diabetes Diagnosis		No Diabetes Diagnosis		
		*n*	%	*n*	%	*p*-Value
Total (N)		587,015	(14.0)	3,617,991	(86.0)	
IFG	No	362,128	(11.4)	2,817,119	(88.6)	<0.0001
	Yes	224,887	(21.9)	800,872	(78.1)	
Age group		48.0 ± 12.7		38.8 ± 11.6		<0.0001
	18–29	49,669	(5.1)	927,995	(94.9)	
	30–39	107,231	(8.6)	1,147,078	(91.5)	
	40–49	169,599	(15.7)	909,147	(84.3)	
	50–59	150,062	(25.6)	435,725	(74.4)	
	60–69	82,528	(34.7)	155,088	(65.3)	
	≥70	27,926	(39.4)	42,958	(60.6)	
Gender	Male	426,674	(14.1)	2,592,429	(85.9)	<0.0001
	Female	160,341	(13.5)	1,025,562	(86.5)	
Household income	1	96,040	(14.8)	551,587	(85.2)	<0.0001
	2	94,698	(12.7)	652,501	(87.3)	
	3	116,804	(12.6)	811,878	(87.4)	
	4	136,771	(13.8)	853,139	(86.2)	
	5	142,702	(16.0)	748,886	(84.0)	
Hypertension	No	484,769	(12.4)	3,416,387	(87.6)	<0.0001
	Yes	102,246	(33.7)	201,604	(66.4)	
Systolic blood pressure	<120	182,114	(10.6)	1,538,074	(89.4)	<0.0001
	120–139	304,390	(14.6)	1,779,601	(85.4)	
	≥140	100,511	(25.1)	300,316	(74.9)	
Diastolic blood pressure	<80	264,121	(11.6)	2,015,260	(88.4)	<0.0001
	80–89	230,515	(15.1)	1,292,284	(84.9)	
	≥90	92,379	(22.9)	310,447	(77.1)	
Dyslipidemia	No	562,230	(13.7)	3,546,414	(86.3)	<0.0001
	Yes	24,785	(25.7)	71,577	(74.3)	
Triglyceride	<150	354,226	(12.1)	2,572,384	(87.9)	<0.0001
	≥150	232,789	(18.2)	1,045,607	(81.8)	
HDL	M < 40, F < 50	103,175	(17.4)	491,130	(82.6)	<0.0001
	M ≥ 40, F ≥ 50	483,840	(13.4)	3,126,861	(86.6)	
BMI		24.5 ± 3.3		23.4 ± 3.1		<0.0001
Waist circumference	M < 90, F < 85	433,978	(12.3)	3,090,889	(87.7)	<0.0001
	M ≥ 90, F ≥ 85	153,037	(22.5)	527,102	(77.5)	
Metabolic syndrome	No	300,847	(18.3)	1,342,253	(81.7)	<0.0001
	Yes	286,168	(11.2)	2,275,738	(88.8)	
Current smoking	No	367,954	(14.5)	2,172,944	(85.5)	<0.0001
	Yes	219,061	(13.2)	1,445,047	(86.8)	
High-risk drinking	No	492,266	(13.5)	3,153,278	(86.5)	<0.0001
	Yes	94,749	(16.9)	464,713	(83.1)	
Proper exercise	No	468,582	(13.6)	2,984,031	(86.4)	<0.0001
	Yes	118,433	(15.7)	633,960	(84.3)	

**Table 4 ijerph-19-00123-t004:** Risk factors for the progression from non-diabetes to diabetes.

Variables		Hazard * Ratio	95% Lower CI	95% Upper CI	*p*-Value
IFG	No	1.000			
	Yes	1.552	1.542	1.563	<0.0001
Age group	18–29	1.000			
	30–39	1.509	1.492	1.525	<0.0001
	40–49	2.724	2.696	2.752	<0.0001
	50–59	4.382	4.336	4.429	<0.0001
	60–69	6.047	5.977	6.119	<0.0001
	≥70	7.504	7.389	7.620	<0.0001
Gender	Male	1.000			
	Female	1.198	1.190	1.206	<0.0001
Household income	1	1.000			
	2	1.003	0.994	1.012	0.5275
	3	1.005	0.997	1.014	0.2438
	4	1.008	0.999	1.016	0.0732
	5	0.931	0.924	0.939	<0.0001
Hypertension	No	1.000			
	Yes	1.443	1.433	1.454	<0.0001
Systolic blood pressure	<120	1.000			
	120–139	1.051	1.043	1.059	<0.0001
	≥140	1.176	1.162	1.189	<0.0001
Diastolic blood pressure	<80	1.000			
	80–89	1.023	1.016	1.030	<0.0001
	≥90	1.082	1.070	1.094	<0.0001
Dyslipidemia	No	1.000			
	Yes	1.256	1.239	1.272	<0.0001
Triglyceride	<150	1.000			
	≥150	1.203	1.195	1.212	<0.0001
HDL	M < 40, F < 50	1.000			
	M ≥ 40, F ≥ 50	0.937	0.930	0.945	<0.0001
BMI		1.048	1.046	1.049	<0.0001
Waist circumference	M < 90, F < 85	1.000			
	M ≥ 90, F ≥ 85	1.183	1.173	1.193	<0.0001
Metabolic syndrome	No	1.000			
	Yes	1.040	1.030	1.049	<0.0001
Current smoking	No	1.000			
	Yes	1.082	1.075	1.088	<0.0001
High-risk drinking	No	1.000			
	Yes	1.140	1.132	1.148	<0.0001
Proper exercise	No	1.000			
	Yes	0.999	0.993	1.005	0.7664

* Adjusted for IFG, age group, gender, household income, hypertension, systolic blood pressure, diastolic blood pressure, dyslipidemia, triglyceride, HDL, BMI, waist circumference, metabolic syndrome, current smoking, high-risk drinking, and proper exercise.

**Table 5 ijerph-19-00123-t005:** Risk factors for the progression from non-diabetes to diabetes from age 40.

Variables		Hazard * Ratio	95% Lower CI	95% Upper CI	*p*-Value
IFG	No	1.000			
	Yes	1.521	1.509	1.533	<0.0001
Age		1.040	1.039	1.040	<0.0001
Gender	Male	1.000			
	Female	1.145	1.137	1.154	<0.0001
Household income	1	1.000			
	2	1.001	0.990	1.011	0.8834
	3	1.000	0.990	1.011	0.9368
	4	0.998	0.989	1.008	0.718
	5	0.908	0.900	0.917	<0.0001
Hypertension	No	1.000			
	Yes	1.394	1.383	1.405	<0.0001
Systolic blood pressure	<120	1.000			
	120–139	1.051	1.042	1.060	<0.0001
	≥140	1.134	1.120	1.149	<0.0001
Diastolic blood pressure	<80	1.000			
	80–89	1.021	1.013	1.029	<0.0001
	≥90	1.068	1.054	1.081	<0.0001
Dyslipidemia	No	1.000			
	Yes	1.248	1.231	1.266	<0.0001
Triglyceride	<150	1.000			
	≥150	1.176	1.167	1.186	<0.0001
HDL	M < 40, F < 50	1.000			
	M ≥ 40, F ≥ 50	0.943	0.935	0.952	<0.0001
BMI		1.042	1.041	1.043	<0.0001
Waist circumference	M < 90, F < 85	1.000			
	M ≥ 90, F ≥ 85	1.145	1.135	1.156	<0.0001
Metabolic syndrome	No	1.000			
	Yes	1.023	1.013	1.034	<0.0001
Current smoking	No	1.000			
	Yes	1.103	1.096	1.111	<0.0001
High-risk drinking	No	1.000			
	Yes	1.131	1.122	1.141	<0.0001
Proper exercise	No	1.000			
	Yes	0.996	0.988	1.003	0.2366

* Adjusted for IFG, age, gender, household income, hypertension, systolic blood pressure, diastolic blood pressure, dyslipidemia, triglyceride, HDL, BMI, waist circumference, metabolic syndrome, current smoking, high-risk drinking, and proper exercise.

## Data Availability

The dataset may not be taken out of the NHIS according to the policy of the National Health Insurance Service of Korea. The data can be accessed on the National Health Insurance Data Sharing Service homepage of the NHIS (https://nhiss.nhis.or.kr/bd/ab/bdaba000eng.do (accessed on 27 October 2021)). Applications to use the NHIS-HEALS data will be reviewed by the inquiry committee of research support. Once approved, raw data will be provided to the applicant for a fee. Although the datasets are coded in English and numbers, not in Korean (Hangul), the use of individual data is allowed only for Korean researchers. Nevertheless, it would be possible for researchers outside the country to gain access to the data by conducting a joint study with Korean researchers.

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
