# Peer review of "Incidence and Risk Factors for Progression to Diabetes Mellitus: A Retrospective Cohort Study"

_ijerph, 2021, doi:10.3390/ijerph19010123_

Round 1

Reviewer 1 Report

The paper by Hyun et al is aimed at assessing the incidence and risk factors that contribute towards the progression to diabetes mellitus (DM) in a seven-year follow-up study of non-diabetic Korean population. This topic is interesting from an epidemiologic perspective because it may help to identify the risk factors which may be influencing the development of Diabetes Mellitus in a healthy population from the moment of its inclusion in the database. The identification of these factors may be crucial to design preventive measures that might avoid the development of diabetes.

However, the paper has several methodological concerns that should be resolved and explained.

Major Concerns

Title and abstract: The author`s should Indicate the study’s design in the title or the abstract. The background is the objective of the study.

Introduction:

From line 58 to 61. I suggest authors to clearly define the objective/s of this study.

The introduction states that impaired glucose tolerance (IGT), either alone or in combination with IFG, are associated with a high risk of incident DM. Why IGT has not been included as a variable in this study? If this variable has not been included in the study, perhaps the introduction should describe the effect of IFG in the development of diabetes, instead of using the concept prediabetes that includes impaired fasting glucose (IFG) and impaired glucose tolerance (IGT).

Material and Methods

Study design and data source

I suggest indicating which main variables are included in the HEALS and NHIS databases. How could the authors have access to the data? Are these data published elsewhere? This information should be added.

Why the follow-up finished in 2015?

Variable definitions

I suggest entitling this section as Variables.

Line 91: Household income. What is the meaning of each quintile?

Line 100: What is the meaning of the acronym FPG?

Line 102-103: I suggest reviewing this criterion as current smoking: those who had smoked more than 100 cigarettes in their lives or were currently smoking.

Other definition states that: “Currently smoking’ refers to someone who has smoked more than 100 cigarettes (including hand rolled cigarettes, cigars, cigarillos etc) in their lifetime and has smoked in the last 28 days”.

I would point out that those who had smoked more than 100 cigarettes in their lifetime and they no longer smoke, can be considered ex-smokers.

Results

Baseline demographics

The authors state that 71.8% of the sample were male. This proportion should be explained because it can’t be an accurate representation of Korean population.

Line 139-141: I suggest clarifying the percentages per sexes, because it is confusing. Par example: “Metabolic syndrome was detected in 60.9% of the total: 54.3% male, and 77.9% female participants”. My suggestion: Of male participants, 54.3% has metabolic syndrome and 77.9% of female.

Why haven’t you described the variable proper exercise by sex? I recommend that you maintain the same criteria in the description of the results of each variable.

Discussion

Line 182: the place of the study should be added.

Line 215-216: the authors state: “The proportion of pre-diabetic subjects at the start of the observation 215 was 21.7%, which is lower than the 24.4% in the present study” Is it possible to define people with IFG elevated as prediabetic? The definition of prediabetes in line 37 includes IGT, and this variable has not been measured. If the criteria of prediabetes can be used with those people who have IGT or IFG elevated, the definition should be clearly formulated.

I suggest commenting the result of metabolic syndrome and the risk of developing diabetes. Can you explain why 18.3% of participants without metabolic syndrome at baseline developed diabetes compared with 11.2% diagnosed with metabolic syndrome.

Reviewer 2 Report

Study subjects: Can missing data regarding smoking/drinking/exercise cause selection bias? Please discuss.

Outcome variables: Was HbA1c used to make the diagnosis of diabetes? If yes, in how many participants. Did you know the blood glucose level oder HbA1c at the time of diagnosis of DM? If Hba1c was used to diagnose DM, was an age related referece range of HbA1c applied or was the cut of of >=6.5% used for people of all ages? The latter could lead to overdiagnosis.

The German Diabetes Assiciation and the German Association of Clinical Chemistry recommend in theire guideline 2019 to be carefule with using HbA1c for diagnosis in elderly people (Petersmann A doi: 10.1055/a-1018-9078). They write “also the increase of the HbA1c which can be an absolute 0.4–0.7 % (4–8 mmol/mol Hb), which is independent of diabetes and comes with age, restricts the use of HbA1c for the diagnosis of diabetes especially in the range below 53 mmol/mol Hb (7.0 %).” The basis for this restriction of the use of Hba1c for diagnosis diabetes in the elderly are two German population studies (Masuch A: doi: 10.1186/s12902-019-0338-7, Roth J doi: 10.1055/s-0042-105440). The German Disease Management Programme in Northrine shows one third of people with diabetes type 2 have HbA1c below 6.5% without any blood glucose lowering drug. Six years later the same people still have  an HbA1c below 6.5% without blood glucose lowering drug. So it looks like these people have no diabetes. Source table 4.10 https://www.zi-dmp.de/Files/QSB17_Tabellenband_V1b.pdf

Please discuss potential influence of dabetes independent HbA1c increase with age on misdiagnosis of DM .

Exercise: could there be a mistake with the definition of intensive and moderate exercise in chapter 2.3.4.?

Baseline demographics;  71.8%  of the participatns were male. Is this representative for Korean population? If no, would could that mean for selection bias?

Author Response

Reviewer 2

Study subjects: Can missing data regarding smoking/drinking/exercise cause selection bias? Please discuss.

Outcome variables: Was HbA1c used to make the diagnosis of diabetes? If yes, in how many participants. Did you know the blood glucose level oder HbA1c at the time of diagnosis of DM? If Hba1c was used to diagnose DM, was an age related referece range of HbA1c applied or was the cut of of >=6.5% used for people of all ages? The latter could lead to overdiagnosis.

The German Diabetes Assiciation and the German Association of Clinical Chemistry recommend in theire guideline 2019 to be carefule with using HbA1c for diagnosis in elderly people (Petersmann A doi: 10.1055/a-1018-9078). They write “also the increase of the HbA1c which can be an absolute 0.4–0.7 % (4–8 mmol/mol Hb), which is independent of diabetes and comes with age, restricts the use of HbA1c for the diagnosis of diabetes especially in the range below 53 mmol/mol Hb (7.0 %).” The basis for this restriction of the use of Hba1c for diagnosis diabetes in the elderly are two German population studies (Masuch A: doi: 10.1186/s12902-019-0338-7, Roth J doi: 10.1055/s-0042-105440). The German Disease Management Programme in Northrine shows one third of people with diabetes type 2 have HbA1c below 6.5% without any blood glucose lowering drug. Six years later the same people still have  an HbA1c below 6.5% without blood glucose lowering drug. So it looks like these people have no diabetes. Source table 4.10 https://www.zi-dmp.de/Files/QSB17_Tabellenband_V1b.pdf

Please discuss potential influence of dabetes independent HbA1c increase with age on misdiagnosis of DM .

Exercise: could there be a mistake with the definition of intensive and moderate exercise in chapter 2.3.4.?

After)

This study had some limitations. First, the HEALS and NHIS databases are second-ary databases that are not planned and collected according to specific research objectives, thereby limiting the validity of the research results. This secondary data is also a limita-tion that cannot be verified due to missing laboratory test results. On the other hand, the use of a secondary database can include large-scale research participants, which makes it easier to generalize the results of the study. Second, there are advantages, such as recall bias and selective reporting reduction. Second, due to data limitations, the changes in health status and health behaviors during the follow-up period could not be analyzed. Despite these limitations, these findings are useful due to the use of a large national pop-ulation-based sample. The findings from this study provide comprehensive epidemiolog-ical information on diagnosed IFG and T2 DM in ROK. Third, we excluded subjects with-out information on smoking, drinking, and exercise in the subject selection process. How-ever, it was an inevitable choice because health behavior information is important to find the risk factors for conversion to diabetes.

 Baseline demographics;  71.8%  of the participatns were male. Is this representative for Korean population? If no, would could that mean for selection bias?

After)

3.1. Baseline demographics

Of the 4,205,006 participants analyzed, the mean age was 40.1 ± 12.2 years, with 71.8% being male. This is because male is more likely to take the national health screening [20].

Reviewer 3 Report

Reviewed article is intersting and important. But it needs correction of References. List of References should be according to instruction for authors. For example. If there are more than ten authors, only max 10 authors can be listed (Author 1, Author 2..... Author 10, et al) - see for exammple Ref. 1.  The names of journals also need changes. For example: Ref. 1. It is "Diabetes research and clinical practice" It should be "Diabetes Res Clin Pract", however in some cases, names of journals are used correct (for example Ref. 29, 30). 

Author Response

Reviewer 3

Reviewed article is intersting and important. But it needs correction of References. List of References should be according to instruction for authors. For example. If there are more than ten authors, only max 10 authors can be listed (Author 1, Author 2..... Author 10, et al) - see for exammple Ref. 1.  The names of journals also need changes. For example: Ref. 1. It is "Diabetes research and clinical practice" It should be "Diabetes Res Clin Pract", however in some cases, names of journals are used correct (for example Ref. 29, 30). 

Response) We used the endnote style of “International Journal of Environmental Research in Public Health”. We also updated outdated web page reference to the latest version. 

Reviewer 4 Report

Kyung Hyun_Incidence and risk factors DM_ijerph_2021

I commend the authors on the completion of this manuscript. Overall it is well designed and on an important topic. I have a few concerns highlighted below.

English language and style must be checked especially in discussion section.

Abstract

Line 25: according to the revision of the results you have to include BMI and waist circumference.

Conclusions: Consider syntax review to: “…, but the incidence of progression to DM was 22.8 per 1000 person-year, higher than previously reported in Korea.”

Introduction

Materials and Methods

Line 68: “HEALS is conducted for health promotion and disease prevention, including DM, for all insured persons.” Has HEALS the same population registry than NHIS? It must be stated in this sentence.

Line 75: Correct to: “The eligible target subjects were selected…”

Line 78: Explain the acronyms, it is the first time they appear. Please, specify the condition referred by the codes.

Line 80: Correct to: “Finally, a 4,205,006 target subject were selected…”

Line 84: It is Figure 2, correct please.

Line 116: “…using survival analysis” change to “Cox proportional-hazards model”

Line 121: what other events apart from follow-up or death were considered?

Line 127: “and expected survival probabilities” correct to “and expected risk probabilities”. “The adjusted HRs” change to “The adjusted hazard ratios (HRs)…”

  1. Results

Table 2: can you explain me please what is the calculation for the column Person-Year? I have not understood the calculation.

Table 4: What is the meaning of * in household income in table 4?

Line 167: Why do you omit mentioning in the text the significative HR referring to Household income, BMI and Waist circumference?

  1. Discussion

Line 191. “On the other hand, the conversion rates were 22.1 and 22.8, respectively, which are slightly lower than the present findings.”. I think you must correct to: “On the other hand, the conversion rate was 22.1, which is slightly lower than the present findings.”

Line 192. “These results can be explained by a comparative study using the data collected in 2001, which is much earlier than 2009, the collection time of the present study.” Please consider syntax review.

Line 217. “…related to the earlier investigation points than in the present study, longer follow-up pe….” I don’t understand why earlier investigation points can explain the higher incidence. Please explain better, that they can explain the lower prediabetes incidence, and the other factors can explain the higher incidence.

Line 225: Please correct to: “…of conditions to understand better the status and trends…”

Line 241: Please consider syntax review to: “Smoking increases the DM risk by affecting body weight (visceral abdominal fat accumulation), peripheral insulin sensitivity, and impaired pancreatic b cell function caused by nicotine exposure”

Line 243: “Various epidemiological studies have demonstrated a risk of DM and its complication [30-33].” Please consider correct to: “Various epidemiological studies have demonstrated a risk for DM and its complication [30-33] linked to smoking.”

  1. Conclusions

Please make the conclusion more concrete according to study results:

For example: “This epidemiological study in the ROK, in a seven-year follow-up period of non-diabetics from among the National Health Examinees, showed that the risk factors are similar to other regions and races. The incidence of progression to DM was 22.8 per 1000 person-year, higher than previous studies in Korea, but much lower than incidence reported from Asian Indian population.”

The next sentences are clinical implications, consider to include it, in the discussion section before limitations.

“Care is needed to prevent DM, and more efforts will be needed to reduce the risk of DM and the number of diabetic patients. In addition, more epidemiological studies in DM are needed to identify and alleviate the disease burden of DM and their complications.”

Round 2

Reviewer 2 Report

The manuscript has improved. However there is one question from my review, which is not answered by the authors:  was HbA1c used for diagnosis of diabetes. If yes, the potential bias of overdiagnosis should be discussed.

Round 3

Reviewer 2 Report

Thank you for your additional information of the potential use and bias by HbA1c for diagnosis of diabetes. I have no further comments.